# The Role of Faith-Based Organizations in Improving Vaccination Confidence & Addressing Vaccination Disparities to Help Improve Vaccine Uptake: A Systematic Review

**DOI:** 10.3390/vaccines11020449

**Published:** 2023-02-15

**Authors:** Uzma Syed, Olivia Kapera, Aparajita Chandrasekhar, Barbara T. Baylor, Adebola Hassan, Marina Magalhães, Farshid Meidany, Inon Schenker, Sarah E. Messiah, Alexandra Bhatti

**Affiliations:** 1South Shore Infectious Diseases and Travel Medicine Consultants, Bayshore, NY 11706, USA; 2School of Public Health, University of Texas Health Science Center, Austin Campus, Austin, TX 78712, USA; 3Center for Pediatric Population Health, UTHealth School of Public Health and Children’s Health System of Texas, Dallas, TX 75027, USA; 4School of Public Health, University of Texas Health Science Center, Dallas Campus, Dallas, TX 75207, USA; 5Caucus on Public Health and the Faith Community, Atlanta, GA 30331, USA; 6Illinois Department of Public Health, Chicago, IL 60612, USA; 7School of Medicine, Stanford University, Stanford, CA 94305, USA; 8Black Pearl Consulting & Research, Leesburg, VA 20175-3012, USA; 9IMPACT, Jerusalem 9107901, Israel; 10Merck & Co., Inc., Rahway, NJ 07065, USA

**Keywords:** faith-based organization, public health, vaccination, immunization, health policy, equity, COVID-19

## Abstract

The COVID-19 pandemic underscored the importance of vaccination to support individual health across the life-course, with vaccination playing a central strategy role in mitigating transmission and disease. This required unprecedented mobilization and coordination across all sectors to meet people where they are, enable equitable access, and build vaccination confidence. A literature search was conducted with combinations of the keywords and variations of vaccination and faith-based organizations (FBOs). Search inclusion criteria were: (1) FBO programs that supported public health emergency efforts, including vaccination efforts as the primary outcome; and (2) articles written in English language. A total of 37 articles met inclusion criteria (*n* = 26 focused on general public health campaigns, *n* = 11 focused on vaccination efforts). The findings related to public health campaigns fell into four themes: FBO’s ability to (1) tailor public health campaigns; (2) mitigate barriers; (3) establish trust; and (4) disseminate and sustain efforts. The findings related to vaccine uptake efforts fell into three themes: (1) pre-pandemic influenza and HPV vaccination efforts, (2) addressing vaccine disparities in minority communities, and (3) enabling COVID-19 vaccination. This review demonstrated that FBOs have a vital role in both public health campaigns and vaccination initiatives to support high vaccine uptake and confidence.

## 1. Introduction

As of February 2023, the COVID-19 pandemic has resulted in over 756 million reported cases, over 6.8 million deaths, and has impacted both adults and children around the globe [1]. A silent crisis has emerged as a result of a convergence of factors related to the pandemic; namely the significant decline in routine vaccinations across the life-course, upending decades of progress in achieving and maintaining high vaccination rates. According to World Health Organization (WHO) and United Nations Children’s Fund (UNICEF) data, approximately 25 million children missed some routine vaccinations and close to 17 million children did not receive a single vaccine in 2020 [2]. This results in communities being placed at risk of vaccine preventable diseases (VPDs), outbreaks, and certain cancers associated with VPDs. The US has recently seen the reemergence of VPDs such as measles, 121 cases occurring in 2022, and polio, underscoring the critical need to maintain high vaccination rates to curb further disease [3,4,5]. Furthermore, there is a disparate impact and a slower recovery of VPDs for the most vulnerable and underserved populations, widening pre-pandemic disparities [1,6,7,8,9,10].

Global declines in vaccination coverage rates across the life-course will take years of recovery to return to pre-pandemic levels—and ultimately achieve global and national vaccination targets [10,11]. Projections estimate that for adolescents in the United States, if every provider saw 15% more patients each month, it could take up to seven years to recover certain vaccinations missed during the COVID-19 pandemic [12]. This calls for a comprehensive approach to ensuring vaccination rates not only increase to pre-pandemic levels, but strengthen immunization programs to achieve high vaccine uptake.

As such, the COVID-19 pandemic has compelled communities and immunization programs to generate creative and sustainable solutions to this public health crisis. Often these solutions are deeply embedded in communities, such as faith-based organizations (FBOs). FBOs and faith-based engagement strategies have been the foundation of many previous collective efforts targeting other infectious diseases and public health emergencies [6,7,13]. FBOs are organizations whose philosophies are driven by certain religious beliefs, often including a social or moral component [14]. These entities have been shown to bring people together for positive purposes and can present powerful agents for health and justice [15,16,17,18]. As religion is a social determinant of population health, it functions through the work of social institutions [16]. Therefore, FBOs are key stakeholders in communities; they present a discernible and trusted public face to communities through acts of leadership and capacity for service to others [15,17]. The CDC workbook defines FBOs as “churches, synagogues, mosques, church sponsored service agencies, and all charitable organizations with religious affiliations”—this broad definition can therefore include nonprofit organizations with a religious affiliation or inspiration [18]. FBOs are driven by a desire to provide health care services, combating the growing unmet health care needs in their community [18]. The social capital effects are of importance in communities with minority and low-socioeconomic groups and elsewhere that social and economic resources are limited, such as vaccinations [17].

The public health imperative of going hyperlocal and engaging FBOs and leveraging faith-based engagement is critical as: (1) the pandemic has worsened long-standing vaccination disparities; (2) lower resource regions and countries share a disproportionate burden of low childhood vaccine uptake; (3) public health challenges to global childhood routine vaccination uptake remain. Yet, there is no synthesis of the current literature on this timely topic. Therefore, we present here the latest summary of evidence on the overall role of FBOs in public health efforts and the role of FBOs in vaccination efforts and vaccine disparities.

## 2. Materials and Methods

### 2.1. Search Strategy

PRISMA guidelines were used as the search framework (Figure 1). A comprehensive search was completed via PUBMED, Web of Science, Science Direct, and COCHRANE. Combinations of the keywords ‘vaccination’ or ‘immunization’, ‘COVID-19 vaccine’, ‘childhood vaccination’ or ‘routine vaccination’ and ‘faith-based organization’ or ‘faith-based’ were used as search terms. The publication dates of interest were limited to 1 January 2002 through 31 December 2021. The search was run across all databases during January–March 2022. See Appendix A for PRISMA Diagram and additional details regarding the search strategy; Appendix A for sample of quality assessment tool for systematic reviews and meta-analyses by the National Heart, Lung, and Blood Institute; Appendix A for sample of quality assessment tool for observational cohort and cross-sectional studies by the Na-tional Heart, Lung, and Blood Institute; Appendix A for sample of quality assessment tool for controlled intervention studies by the National Heart, Lung, and Blood Institute; and Appendix A and sample of quality assessment tool for case series studies by the National Heart, Lung, and Blood Institute.

### 2.2. Study Selection

Studies were included if they met the following criteria: (1) FBO programs that supported public health emergency efforts, including vaccination efforts, both routine and COVID-19, as the primary outcome; and (2) articles written in English language.

Literature was excluded: (1) stand-alone abstracts; (2) unpublished articles. Duplicates were manually identified and excluded.

For the context of the study, FBOs were defined as entities associated with or inspired by religion or religious beliefs. Public health campaigns were understood to be an effort to motivate a defined public to participate in behaviors that will improve health or withhold from behaviors that are identified as unhealthy.

### 2.3. Data Extraction

Studies were selected based on a theoretical framework of FBOs and public health interventions (Figure 2) [15,16,17,18,19]. The theoretical framework proposes FBOs are able to enact public health interventions through four themes: (1) tailor public health campaigns, (2) ability to manage barriers and challenges, (3) dissemination and sustainability, and (4) establishing a community of trust (Table 1). Similarly, we examined the role of FBOs in vaccine uptake specific efforts through three themes: (1) pre-pandemic influenza and HPV vaccination uptake efforts, (2) addressing vaccine disparities in ethnic minority communities, and (3) addressing recent COVID-19 vaccination efforts (Table 2). The themes listed were developed by identifying the overarching goal of each study and subsequently grouping them based on corresponding goals.

Studies were stratified by (1) any FBOs efforts to improve public health interventions (Table 1); (2) FBOs efforts to improve vaccine uptake in general (Table 2); and (3) FBOs efforts to improve COVID-19 vaccine uptake (also Table 2).

The papers were also organized according to: origin of data collection, purpose, research design, target population, inclusion criteria, summary points, and key points. This helped to map the overall landscape of the literature.

Data extraction was completed by 2 reviewers (O.K. and A.C.). There were no discrepancies between reviewers in terms of data extracted or choice of articles meriting inclusion.

The study did not collect any primary data. The risk of bias in individual studies was analyzed using the Study Quality Assessment Tools developed by the National Heart, Lung, and Blood Institute (NHLBI) [20]. The strength of studies was evaluated based on criteria published by the Oxford Center for Evidence-Based Medicine (OCEBM) [21].

## 3. Results

A total of 37 articles were included in the analysis (Figure 1) [6,7,22,23,24,25,26,27,28,29,30,31,32,33,34,35,36,37,38,39,40,41,42,43,44,45,46,47,48,49,50,51,52,53,54,55]. Results were categorized as “public health interventions”, where interventions were related to addressing public health needs across various topics including mental health and physical activity, then “vaccination interventions” which are those focused specifically on supporting vaccine uptake. Results are presented below in order of those based on public health interventions first, then further refined to those focused on vaccination interventions.

### 3.1. Summary of the Role of Faith-Based Organizations in Public Health Interventions

Table 1 summarizes the articles (*n* = 26) that were deemed relevant or directly related to FBOs and public health interventions. Of the 26 articles, 14 of the studies were conducted in the United States. The findings in context to public health campaigns and efforts in general are summarized in four themes based on the theoretical framework (Figure 2): FBO’s ability to (1) tailor public health campaigns, (2) ability to manage barriers and challenges, (3) dissemination and sustainability, and (4) establishing a community of trust.

#### 3.1.1. Theme 1: Tailored Public Health Campaigns

Leyva et al. [22] concluded that public health campaigns are not one-size-fits-all for FBOs, and there is a need for tailored strategies to enhance community engagement. Due to the involvement in its members’ lives, FBOs can address individual concerns in addition to personalizing the public health campaign, as found in Wasser et al. [23] For utilizing the sense of community within FBOs, Wilson et al. [24] developed a framework for FBOs to create culturally appropriate public health campaigns. Similarly, Blankinship et al. [25] performed a narrative review on faith-based wellness programs in Latino and African American populations and determined FBOs provide culturally appropriate messaging and create community support and self-efficacy, which aid in public health campaigns. Similarly, Sobers et al. [26] showcased FBOs’ use of social support to implement evidence-based self-management intervention for hypertension. Ackerman Gulaid and Kiragu [27] provide ten recommendations for community engagement that highlight the need to actively identify and build on a community’s efforts.

#### 3.1.2. Theme 2: Ability to Manage Barriers and Challenges

The ability to remove barriers that limit access to care for their members is another attribute of the application of FBOs for public health campaigns. In a retrospective chart review of Hispanic farmworker women in Central Florida, Luque et al. [28] found that FBOs aided in the removal of obstacles that prevent uninsured and low-income populations from receiving and accessing care. Tristão Parra et al. [29] reviewed the use of physical activity interventions delivered by FBOs and concluded FBOs create an environment that encourages physical activity and, therefore, can address health disparities. Similarly, in a review of Asian American FBOs in New York City and New Jersey, Kwon et al. [30] determined that FBOs support public health initiatives that give people who have disproportionately poor health outcomes, access to health promotion programs.

Other studies have shown that FBOs provide their members with various coping strategies to address mental health barriers such as prayer, scripture, social support groups, religious services, and aid from religious leaders, showing a potential opportunity for other health-related dialogue such as vaccination. Rayes et al. [31] performed qualitative interviews on Arabic-speaking refugee populations from Muslim majority countries resettling in Europe. The study found that participants with a stronger devotion to faith were more likely to use faith-based coping mechanisms when handling challenges and seeking mental health services. Similarly, Schieffler and Genig [32], after conducting a mini-narrative review on orthodox church members, found that FBOs play a significant role for its people regarding mental health burdens to heal “both soul and body”.

#### 3.1.3. Theme 3: Establishing a Community of Trust

One of the essential aspects of FBOs involvement in public health campaigns is the capability to create a community of trust. Bopp et al. [33] emphasized FBO leaders’ role in distributing health promotion interventions. Members entrust their leaders to shape their spiritual, physical, emotional, and social environments. In a narrative review, Maynard [33] concluded FBOs are a central component in neighborhoods, and participation in a faith community persists in many communities.

FBOs play a crucial role in the daily life of African American faith communities. As explored in Francis and Liverpool [35], the adoption of the public health campaigns by faith community leaders influences the adoption in African American communities. Lancaster et al. [36] performed a systematic review on African American FBOs and concluded FBOs could effectively promote health interventions and behaviors in their communities due to the significance of faith to many African Americans.

By creating a community of trust, FBOs provide an ecosystem of aid. FBOs coordinate and collaborate with health professionals, stakeholders, community members, and social groups. Vitillo et al. [37] concluded from their review of the role and contributions of FBOs, private sectors, and philanthropic partners that joint engagement is required to achieve community success. In a review of the relationship between government and FBOs, Brooks and Koenig [19] found partnerships between government and FBOs may create an environment with the most significant health benefit that could not be accomplished solely. Pfefferbaum et al. [38] found partnerships in public health interventions produce the most significant positive impact.

#### 3.1.4. Theme 4: Dissemination and Sustainability

As explored in a mixed-methods study by Grieve and Oliver [39], FBOs are a recognized, valued stakeholder in society, allowing them to collaborate among those in the public sector and various community members. These partnerships allow for an increase in the public health campaign’s awareness, longevity, and effect. Bopp and Fallon [40] concluded that partnering with FBOs for public health campaigns presents several advantages and prospects for reaching specific populations. Kaczynski et al. [41] performed a group-randomized trial in which they found FBOs can encourage diverse health outcomes. Additionally, Johnston et al. [42] concluded health outcomes could be improved through public health interventions by FBOs.

For a public health campaign to be sustainable, there needs to be buy-in from community members, policymakers, and other relevant decision makers and stakeholders. With FBOs being a trusted source, FBOs can engage members interested in public health campaigns, as seen by Morad et al. [43], which found involvement of FBOs in public health campaigns engages community participation. Sheikhi et al. [7] concluded FBOs provide avenues for partnership with stakeholders and collaboration with healthcare providers, especially mental health professionals. Tadesse Gebremedhin et al. [44] found that committed leadership by policymakers and “buy-in” from crucial stakeholders allows for the involvement of community members at different levels. In a systematic review, DeHaven et al. [45] provided recommendations for FBOs to increase their effectiveness by encouraging partnerships between FBOs and health professionals to evaluate public health interventions, disseminate findings, and focus on fostering relationships with racially and ethnically diverse populations.

### 3.2. Summary of the Role of Faith-Based Organizations in Vaccination Interventions

Table 2 provides a summary of the role of FBOs in vaccine uptake specific efforts. It must be noted, however, that the literature on vaccination efforts related to COVID-19 vaccines is limited, given that the vaccine has only recently become available for children. In general, the 11 included papers [6,46,47,48,49,50,51,52,53,54,55] focused on three main themes: (1) pre-pandemic influenza and HPV vaccine uptake efforts, (2) addressing vaccine disparities in ethnic minority communities, and (3) supporting recent COVID-19 vaccination efforts.

#### 3.2.1. Theme 1: Pre-Pandemic Influenza and HPV Vaccine Uptake Efforts

Zimmerman et al. [46] conducted a pre-post study to assess whether interventions tailored to individual practice sites increased influenza vaccination rates among high-risk children at inner-city health centers over two years. The authors found that FBOs saw the highest rates of vaccination compared to other practices and were associated with the highest rates of children obtaining the influenza vaccine. Similarly, Bond et al. [47] showed that FBOs in New York are uniquely equipped with resources to deliver health promotion programs to community members versus academic and governmental organizations, which resulted in high retention rates.

Kiser and Lovelace [48] assessed a national (US) collaboration between the Interhealth Health Program (IPH) at Emory University, the Department of Health and Human Services Partnership Center, the Centers for Disease Control and Prevention (CDC), and the Association of State and Territorial Health Officials to prevent the spread of 2009 H1N1 and seasonal influenza by leveraging community organizations (FBOs being one of them). The authors found that forming national partnerships is critical in mobilizing local community resources and organizations such as FBOs to improve vaccination efforts among underserved populations.

Regarding HPV vaccine uptake efforts, Lahjiani et al. [49] conducted a qualitative case study in the US to understand community perceptions of the HPV vaccination among both leaders and members of an African Methodist Episcopal (AME) church in Atlanta, Georgia. This study made use of a Social and Behavior Change Communication framework in order to increase the promotion of the HPV vaccination in the future. Focusing on findings related to church leaders, this church-based intervention found that leaders were amenable to (1) having more trust in the healthcare system and (2) viewing the HPV vaccine in a less stigmatized manner, instead promoting uptake of the vaccine among church community members. The authors stated that mistrust in the healthcare system by intervention participants was likely due to unethical treatment of study participants in the Tuskegee Syphilis Study. Conducting this study was important as HPV prevention within this specific church community has never been done before. As a result, this study serves as an important step for leaders of this church community to mobilize and promote the uptake of the HPV vaccine within the AME community. Similarly, Olagoke et al. [50] conducted a cross-sectional study in the US to assess the associations between three domains of religiosity (organizational, non-organizational, and intrinsic) and the intention to obtain HPV information and receive the HPV vaccine. This study found a positive association between the organization domain and the intention to seek HPV information. However, it was also found that information-seeking may not lead to vaccination. This study underscored the importance of engaging FBOs and empowering them to provide parents with accurate information regarding HPV vaccination to increase HPV vaccination rates.

#### 3.2.2. Theme 2: Addressing Vaccine Disparities in Ethnic Minority Communities

Daniels et al. [51] conducted a randomized controlled trial in the US to examine (1) whether church-based vaccine education increased adult utilization of vaccinations in ethnic/minority communities, and (2) if churchgoers who are offered vaccinations in churches have higher vaccination utilization rates compared to non-churchgoers. The target population for this study was African American and Latino adults aged 65 and older living in San Francisco, California. This study showed the value of FBOs in decreasing the disparities in vaccination rates among racial/ethnic communities.

Santibañez et al. [6] discussed how CDC-FBOs collaborations played a key role in the response to pandemic influenza (2009). This vaccination effort focused on ethnic minority communities in the Minneapolis-St. Paul area. The Minnesota Immunization Networking Initiative (MINI) conducted vaccination clinics at places of worship (Churches, a Hindu Temple, mosques, a Buddhist monastery) and provided free influenza vaccinations. The collaboration of MINI with different FBOs helped address barriers to influenza vaccinations (access, mistrust, transportation) among underserved groups.

#### 3.2.3. Theme 3: Supporting Recent COVID-19 Vaccination Efforts

Monson et al. [52] conducted a narrative review that showed how medical-religious partnerships are practical and valuable in mitigating the impact of COVID-19-related disparities in the US (e.g., using FBOs as COVID-19 testing sites and as potential COVID-19 vaccine sites). Dascalu et al. [53] conducted a case study in Romania to discuss the contributions of the Romanian Orthodox Church (ROC) in mitigating the impact of COVID-19; vaccination efforts being one. This study found that the ROC advocated against misinformation regarding both COVID-19 in general as well as the national vaccine campaign.

Rachmawati et al. [54] conducted a qualitative case study in Indonesia to assess the strengths of Indonesia’s two largest Islamic FBOs and the challenges faced while conducting activities to mitigate the impact of COVID-19 nationally. The target population for this study were informants (the heads of the special units of both FBOs, government officials, etc.). An important finding of this study was that the Central Board of one of the FBOs provided support for COVID-19 vaccination implementation. Overall, the collaboration of FBOs with the government aid in mobilizing resources to help reduce the impact of COVID-19 in communities across Indonesia.

In contrast to the ten studies summarized above, it is important to note that Rujis et al. [55] conducted a qualitative study (grounded theory) in the Netherlands to determine the role of Protestant religious leaders in promoting acceptance or refusal of vaccination by members of the church. This study showed that the role of Protestant religious leaders in influencing the acceptance or refusal of vaccination was split into three subgroups—(1) those who do not recognize the need to address vaccination as congregation members already accept it, (2) those who focus on giving congregation members the choice, and (3) those who preach against vaccination. All three subgroups believe vaccination in the Netherlands should continue to be voluntary. Additionally, pastors are unwilling to promote vaccination on behalf of the authorities. Thus, this shows that using FBOs to promote vaccination in the Netherlands may not be effective because Protestant religious leaders are unwilling to promote vaccination and there is a low level of religiosity in the general population.

## 4. Discussion

This timely systematic review summarizes the opportunities for FBOs, built on key themes, to drive public health policy and action to support public health vaccine uptake efforts. Despite the recognized value of FBOs, they remain under-utilized in supporting routine vaccination, resulting in somewhat limited published studies on effectiveness for discreet areas of public health concern, such as vaccination uptake. However, FBOs have played a robust role in COVID-19 vaccine uptake in the US and globally [56,57]. Additionally, the COVID-19 pandemic has underscored the critical importance of going hyper-local to engage community-based organizations such as FBOs to increase vaccine confidence, allay vaccine concerns, and ultimately drive vaccination uptake. As such we also provide promising strategies that are being leveraged now to support COVID-19 vaccination. For instance, the All Dulles Areas Muslim Society (ADAMS) Compassionate Healthcare Network in Virginia is run by an interfaith team and is committed to providing community members with access to the COVID-19 vaccine [58].

Additionally, Faiths4Vaccines is a highly active group of religious leaders from across the United States which was established during the COVID-19 pandemic to promote vaccine equity. The organization is led by religions and medical leaders and has brought together over 1000 faith leaders across diverse communities in the United States to promote vaccine access and equity. They held 13 bi-weekly roundtables to support a knowledge-exchange for best practices, led by those who have utilized their house of worship as a vaccination site. Faiths4Vaccines had regular engagement with the White House Office on Faith based and Neighborhood Partnerships, the White House COVID-19 Task Force, the Center for Disease Control and Health and Human Services, as part of these bi-weekly calls.

Faiths4Vaccines convened the largest interreligious National Summit in the United States focused on vaccine access and uptake. The Summit connected over 800 registered local and national faith groups, medical professionals, and government officials to identify opportunities for faith leaders and institutions to identify mechanisms and collaboration opportunities to advance equitable vaccine acceptance and uptake. In addition, youth for faith initiative was established in June 2021, which consisted of a roundtable discussion showcasing how youths of faith are leading in their communities within the COVID-19 vaccination efforts. An interfaith clinic vaccination site was established in Washington D.C. The clinic hosted at the mosque was part of Washington, D.C.’s “Faith in the Vaccine Initiative”, which has administered more than 4600 vaccines since February 2021. Lastly, as demonstrated by the evidence and promising findings in this review, partnering with FBOs is particularly critical to engage traditionally underserved, marginalized, and hard-to-reach populations, which can help address health and specifically vaccination disparities that have only widened during the COVID-19 pandemic.

Similarly, the Jerusalem Impact Vaccination Initiative convened leaders of the Jewish, Christian, Muslim, Druze, and Bahá’í Faiths of the Holy Land in November 2021, to develop an action plan on improving vaccination coverage and community resilience. Through the President of the State of Israel, this initiative widened its scope to similar dialogues in Germany and Switzerland.

This review found that previous public health efforts that partnered with FBOs fell into four general themes. All of these themes are critically important in the context of barriers to vaccine uptake barriers. While there is a strong evidence base to support the benefits of receiving routine vaccinations across the life-course, there is an equally strong literature base documenting disparities in the acceptance and uptake of vaccines among individuals from racial and ethnic minority populations [59,60].

Vaccine hesitancy had steadily increased in the United States prior to the COVID-19 pandemic and has been amplified further during the pandemic [61]. Vaccine hesitancy is described as one’s confidence in the safety and efficacy of the vaccines, complacency towards vaccine uptake/completion, and convenience in accessing the vaccine. The reasons for hesitancy are varied, but many include the following; (1) concerns about perceived safety, (2) skepticism about the trustworthiness of the source(s) of vaccination recommendations, (3) exposure to misinformation and disinformation, (4) immunization considered a low priority, (5) perceived low risk of illness, (6) limited knowledge and health literacy about the disease, (7) difficulty accessing services, (8) clinician bias, (9) cost, or (10) personal, cultural or religious beliefs discouraging vaccination [62,63]. According to a 2019 national survey conducted by the American Academy of Pediatrics, >1 in 4 parents reported hesitancy about influenza vaccinations [11]. Only 1 in 4 parents believed the influenza vaccine was effective and 1 in 8 parents had concerns about the safety of both influenza and other childhood vaccines.

Studies have further documented a pattern of racial and ethnic minority persons being less likely to receive the influenza vaccination, with socioeconomic and clinician/health care system factors as main predictors of poor uptake. Research on HPV vaccine uptake shows a similar pattern of racial and ethnic minority persons being less likely to initiate or complete the series [64]. A survey conducted by the Public Religion Research Institute conducted in March 2021 found that 36% of Black Protestants and 33% of Hispanic Americans who are vaccine hesitant say one or more faith-based approaches would make them more likely to get vaccinated [13].

The COVID-19 pandemic has illuminated global vaccine uptake disparities that require innovative public health strategies. Utilizing a multipronged approach to enhance vaccination rates with both strategic and creative approaches can aid in advancing science. Indeed, effective measures and trusted resources, such as FBOs, are essential to addressing and reducing misinformation, building community trust, and promoting widespread vaccine dissemination, population level uptake, and adherence to vaccination protocols. Upstream factors (e.g., interpersonal, community, health system, policy), relevant cultural and historical factors associated with individual beliefs, risk perceptions, and behavior across multiple levels (e.g., individual, community, etc.) must all be considered. Importantly, policy guidance is needed to address misinformation, distrust, and hesitancy regarding the uptake of vaccines (e.g., COVID-19, pneumococcal, influenza, hepatitis B, human papillomavirus (HPV)), especially in populations at increased risk for morbidity and mortality due to long-standing systemic health and social inequities and chronic medical conditions.

Notably, we also found that the majority of research exploring the role of FBOs and faith-based engagement, occurred in the US (14 of 26), other countries examined came up singularly: Barbados, Israel, Indonesia, Romania, Germany, Ethiopia, Ghana, Southeast Asia, broadly. Given the disproportionate volume of research focused on efforts in the US, comparing findings across countries based on available evidence is challenging. However, consistently, while the value of FBOs and faith-based engagement is identified particularly in supporting underserved and traditionally marginalized populations, most studies underscored the importance of coordination with ongoing vaccination efforts so as to not undermine or duplicated efforts, but rather enhance and build upon them. No discernable differences were identified between how FBOs are engaged across countries included in the studies that were captured in this review, with the exception of Rujis et al. [56] which focused on the role of Protestant religious leaders in the Netherlands, which found that in this case, engaging Protestant religious leaders to support vaccination efforts may be ineffective as leaders were unwilling to promote vaccination, and there was a low level of religiosity in the general population. This provides important insights moving forward in that the potential impact of FBOs and faith-based engagement in supporting vaccination efforts may be determinant on the local level faith-community’s buy-in and willingness to engage on this issue as well as the level of religiosity in the community.

It is important to acknowledge the limitations in this review. First, the intersection between faith-based organizations and vaccination is complex and multifaceted, spanning the breadth of vaccine education, vaccine hesitancy, health equity, vaccine programming for communities of faith, the role of faith-based organizations in supporting vaccine efforts in communities beyond their own, and others. Moreover, improving vaccination rates in and of itself is a complex subject, with limited generalizability across different contexts (ex. emergency response; primary care; domestic vs. global settings) and unique challenges with uptake of specific vaccinations among different target populations (ex. MMR vs. HPV vs. COVID-19 vaccine). Lastly, the focus on FBOs is, at times, in danger of glossing over racial/ethnic inequalities in health, when the intersection of religion and race/ethnicity is important to consider in explicit detail.

This review did identify important gaps in the literature. First, the importance of the role of FBOs in vaccination efforts among pediatric populations has received little attention in the literature. Additionally, there is a lack of evidence and outcome-based research on the role of FBOs in vaccination uptake efforts, particularly in assessing the effectiveness of health programs, community participation, and project continuation. Similarly, the narrow field of literature is dominated by studies of pandemic and non-pandemic influenza vaccination in the US. Therefore, there is a lack of understanding of how FBOs affect the implementation of programs in various settings and policy contexts. Future research should also further examine any differences or commonalities in the role of FBO and faith-based engagement across high income and low- and middle-income countries.

In summary, the global scale of disruptions to routine vaccination due to the COVID-19 pandemic highlights the risk of potential VPDs outbreaks in the future and the urgent need for “catch-up vaccination strategy implementation for vulnerable populations and ensuring vaccine coverage equity and health system resilience” [65]. FBOs can and should be a part of this solution based on their solid track record of previous partnerships that have led to public health successes.

## 5. Conclusions

This review found that generally FBOs play a vital role in global vaccination efforts, represent significant potential for effective public health and vaccination campaigns, and can help improve vaccination efforts, particularly within traditionally underserved populations. Based on the literature, it is recommended that FBOs be positioned to build trust with communities, activate existing networks, draw on expertise, involve community partners, and resolve conflict. As trusted members of communities, FBOs allow public health interventions to be modified so that individuals are receptive to the messaging. With an established relationship of trust, individuals will be inclined to participate in the vaccination campaigns introduced by their FBOs. Vaccination and policy stakeholders should explore engaging with, enabling, and resourcing FBOs in implementing local vaccination programs and practices to achieve and maintain high vaccination coverage rates.

## Figures and Tables

**Figure 1 vaccines-11-00449-f001:**
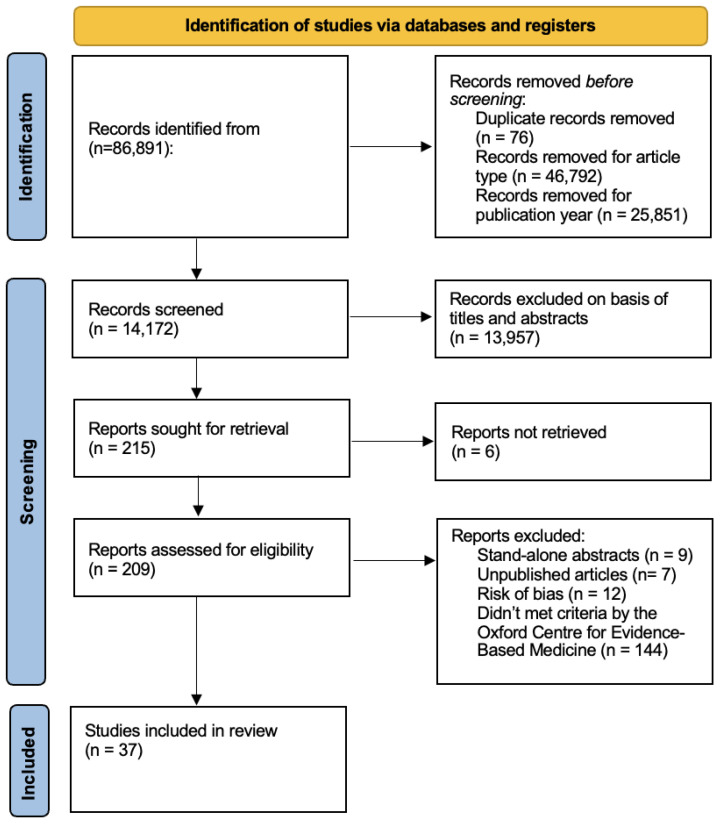
Method flow-chart.

**Figure 2 vaccines-11-00449-f002:**
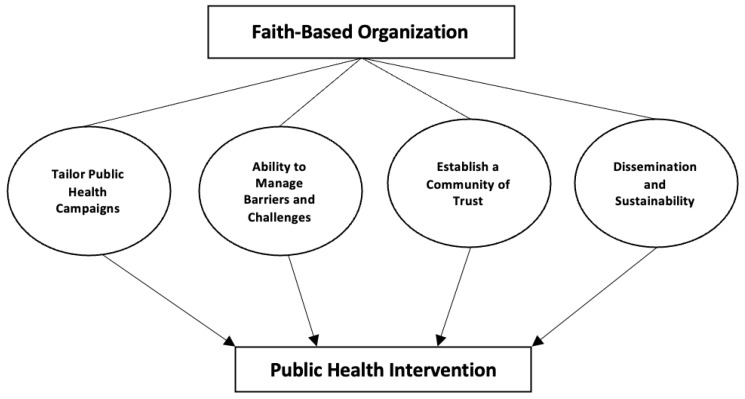
Theoretical Framework of the Role of Faith-Based Organization.

**Table 1 vaccines-11-00449-t001:** Summary of Published Studies Focused on the Role of Faith Based Organizations in Public Health Campaigns and Efforts.

Authors	Origin	Purpose	Research Design	Target Population	Inclusion Criteria	Summary of Findings	Key Points
Theme 1: Tailored Public Health Campaigns							
**Sobers (2021)**	Barbados	Evaluate the effect of an online chronic disease self-management program (CDSMP) plus medication adherence tools on systolic blood pressure (SBP) (primary aim) and seek to understand the barriers and facilitators to implementation of this modified CDSMP in faith-based organizations (FBOs) (secondary aim)	Unblinded cluster randomized trial	Attendees of faith-based organizations	Persons ages 35–70 years; a previous diagnosis of hypertension or currently on antihypertensive therapy and the occurrence of two or more blood pressure readings above 130 mm Hg (systolic) or 80 mm Hg (diastolic) on the day of recruitment; persons not known to have hypertension but who have two or more blood pressure readings at or above 130 mm Hg (systolic) or 80 mm Hg (diastolic) on two recruitment days at least 1 week apart	The study leverages the social support mechanisms existing within faith-based organizations (FBOs) to disseminate an evidence-based self-management intervention for hypertension. Findings on factors impacting implementation should be transferable to other small island developing states/territories with close-knit vibrant FBOs. Engagement of FBOs is likely to under-represent men, as FBO membership in our trial sites consist of more than 70% women.	FBOs are able to interact with the community to deliver an evidenced-based intervention that has been modified for cultural relevance and to emphasize medication adherence
**Blankinship (2021)**	USA	Investigates the outcomes of faith-based wellness programs on Latino and African American populations with respect to general health and wellness, obesity management, DM type 2, and hypertension	Narrative review	Faith-based wellness programs in Latino and African American population		Perceived authority of faith community nurses, faith leaders, and accountability and encouragement provided by faith communities are critical. Long-term behavior change is positively affected by elements faith-based organizations can provide cultural appropriateness, community support, and self-efficacy	Faith-based organizations can provide cultural appropriateness, community support, and self-efficacy which can aid in health messaging and acquiring healthy behaviors
**Wasser (2018)**	Israel	Evaluate live kidney donations facilitated by the Matnat Chaim organization and referred to Israel transplant centers, since the organization’s inception in 2009, was performed and compared to published data from the Israel Ministry of Health	Retrospective review	Live donor transplants in Israel	Individuals referred by Matnat Chaim organization for kidney transplantation after the establishment of the organization in February 2009 until December 2016	The success of an Israel community organization in the promotion of kidney transplantation may serve as a model for other religious and non-religious communities worldwide	Faith-based community organizations help their community members by personalizing processes and addressing individual’s concerns
**Leyva (2017)**	USA	Explore the role of faith-based organizations (FBOs) in extending the reach of EBIs to promote cancer control among ethnic minority groups, such as Latinos	Qualitative interviews	Faith-based organizations	Community stakeholders with diverse and complementary expertise in capacity building, FBOs, community partnerships, and leadership within Catholic parishes, and who had affiliations with Latino-serving community organizations	Capacity building intervention may be needed to facilitate adoption and implementation of cancer control EBI in parish settings. Notable barriers exist, including lack of knowledge of existing EBIs as well as limited financial resources and paid outreach personnel	There is a need for tailored strategies rather than a one size fits all approach for capacity enhancement of local parishes
**Wilson (2013)**	USA (FL)	A comparative effectiveness trial supporting the IDM process and systematic literature reviews to add to CERED’s goal of providing evidence to eliminate cancer health disparities	Effectiveness trial and literature review	Black men in the Tampa Bay area	Self-identification as African American, never having been diagnosed with prostate cancer, and being between the age of 40 and 70 years	The project provides a potential framework for the development of culturally appropriate activities and resources to improve IDM for prostate cancer screening among Black men	Utilization of community health workers who are trained to work on multiple levels to enhance knowledge and awareness, increase interpersonal communication, and support organizational involvement and engagement
**Ackerman Gulaid (2012)**		Summarizes the results of a review commissioned by UNAIDS to help inform stakeholders on promising practices in community engagement to accelerate progress towards these ambitious goals	Literature review and key informant interviews			Ten recommendations on community engagement are offered for ending vertical transmission and enhancing the health of mothers and families: (1) expand the frontline health workforce, (2) increase engagement with community- and faith based organizations, (3) engage communities in program monitoring and accountability, (4) promote community-driven social and behavior change communication including grassroots campaigns and dialogues, (5) expand peer support, (6) empower communities to address program barriers, (7) support community activism for political commitment, (8) share tools for community engagement, (9) develop better indicators for community involvement and (10) conduct cost analyses of various community engagement strategies	As programs expand, care should be taken to support and not to undermine work that communities are already doing, but rather to actively identify and build on such efforts. There should be sustained engagement and unique inputs of various communities, from small informal groups at the grassroots level right up to global coalitions.
**Theme 2: Ability to Manage Barriers and Challenges**							
**Rayes (2021)**	Germany	Explore the manifestation of faith-based coping strategies among Arabic-speaking refugee adults seeking mental healthcare services in Berlin, Germany and explore how favorable faith-based coping strategies can be optimized from a mental health service-delivery and broader integration perspective	Qualitative interviews	Arabic-speaking refugee populations from Muslim majority countries resettling in Europe	Arabic-speaking refugee adults seeking mental health services at the Charité Universitaetsmedizin in Berlin	Participants’ dynamic relationship with their faith following their arrival to Germany played a direct role in how faith-based coping methods were or were not utilized when experiencing mental health symptoms	Faith and faith practices play a significant role in the mental health and integration of people into society
**Schieffler (2021)**	USA	Identify ways in which other faith-based organizations have studied mental health in their own communities	Narrative mini review	Orthodox Church members	Articles that addressed aspects of immigrant and minority groups which closely resembles the experience of Orthodox Christians in the USA given the ethnic preferences and insularity of many parishes	There are resources available for Church Leadership if that leadership takes the time to explore those options. Given similarities that exist between minority and immigrant-based faith-based populations and Orthodox communities, a planning framework is suggested to improve an Orthodox response post-pandemic	Through local partnerships and internal organizations, faith-based organizations can play a sizable and dramatic role in the recovery of its people from the mental health burden of recent events
**Tristão Parra (2018)**	USA	Review and assess the effectiveness of physical activity interventions delivered in faith-based organizations		Members of faith-based organizations	Randomized and nonrandomized controlled trials investigating physical activity interventions for adults delivered in faith-based organizations	Interventions delivered in faith-based organizations increased physical activity and positively influenced measures of health and fitness in participants	Faith-based organizations are promising settings to promote physical activity, consequently addressing health disparities
**Kwon (2017)**	USA	Present baseline results and describes the cultural adaptation and implementation process of the REACH FAR program across diverse FBOs and religious denominations serving Asian American subgroups	Review	Asian American faith-based organizations in New York City and New Jersey		Most Asian American FBOs lack nutrition policies and present prime opportunities for evidence-based multi-level interventions. REACH FAR presents a promising health promotion implementation program that may result in significant community reach	Faith-based health promotion initiatives can provide access to people who disproportionately experience poor health outcomes and are not reached by mainstream traditional health care channels
**Luque (2011)**	USA (FL)	Describe clinical outcomes of an outreach partnership between a cancer center and a faith-based outreach clinic offering gynecologic screening services to increase cervical cancer screening adherence in primarily Hispanic farmworker women	Retrospective chart review	Hispanic farmworker women in central Florida	Women who received cervical cancer screening in well-women care at CMMS from January 2003 to October 2006	The partnership between academic medical centers and faith-based community was successful in increasing cervical cancer screening adherence in the medically underserved population	Faith-based clinical outreaches aid in the removal of barriers that prevent uninsured and low-income populations from receiving and accessing care
**Theme 3: Public Health Intervention Dissemination and Sustainability**							
**Tadesse Gebremedhin (2021)**	Ethiopia	Examine the policies, delivery models, and Lessons learned from Ethiopia’s experience in integrating MH/SA services into primary care through participant observation and literature review	Literature review and participant observation			Mental Health and Substance Abuse tasks are performed by multiple levels of health providers who receive training and periodic refresher training. These tasks are covered at each level of health sector, local, district, and national. Basic MH/SA services are therefore available to the public at large, no matter where in the country they live and what type of health facility or provider is closest.	The importance of committed leadership by policymakers, “buy-in” by critical stakeholders, and calibrated integration of services enable task-sharing across different levels health-care providers, and provision of adequate supervision and mentorship to trainees to ensure proper acquisition of skills for delivery of quality care. Program monitoring and the mobilization of adequate financial and human resources are essential factors to ensure quality and sustainability of primary health care
**Sheikhi (2020)**		Investigate the role of religious institutions in disasters management	Systematic review		The papers from a broad range of disciplines related to keywords	Religious institutions contribute to response and recovery phases of disasters, although these services are valuable, but the great potential of these groups should also be recruited to participate in preparedness and mitigation efforts as part of disasters cycle. Coordination and collaboration of all stakeholders is essential in this way	Religious institution provides avenues for partnership with stakeholders, help collaborate and communicate with mental health professionals, display a unity front for information and messages, deal with disaster in old and new approaches, and assist with barriers and challenges
**Johnston (2018)**	USA	Describe the processes utilized to design and implement an initiative to increase capacity for laity-led comprehensive health ministry among Kansas United Methodist Church congregations and to share the key elements of the initiative	Review	Laity-led health ministry teams			Health outcomes can be improved though faith-based health interventions
**Grieve (2018)**	Ghana	Geospatial mapping of the Ghanian FBNP health sector to provide a visual representation of changes in the spatial footprint	Mixed-methods study	FBNP in Ghana		FBNPs have had a long-standing role in the provision of health services and remain an asset within national health systems in Ghana and sub-Saharan Africa more broadly. Collaboration between the public sector and such non-state providers, drawing on the comparative strengths and resources of FBNPs and focusing on whole system strengthening, is essential for the achievement of UHC	Resilient health system is more likely to be achieved if a whole systems perspective is taken, that is inclusive of and capitalizes on the strengths and resources that continue to be offered by FBNPs and other NSPs in many LMICs
**Kaczynski (2018)**	USA	To describe development and reliability testing of a novel tool to evaluate the physical environment of faith-based settings pertaining to opportunities for physical activity (PA) and healthy eating (HE)	Group-randomized trial	Churches in a rural and medically underserved community, with a larger population of Black/African American residents relative to the state as a whole	Churches located in the target county, had a membership of at least 20 individuals, and were willing to be randomized to either early or delayed training	The tool proved reliable and efficient for assessing church environments and identifying potential intervention points	Through faith-based partnerships, the church environments influence diverse health outcomes
**Morad (2015)**	Southeast Asia	Review of the experience of selected countries in Southeast Asia, where such support, especially from philanthropic organizations, has enhanced the treatment of patients with chronic kidney disease (CKD)	Review	Philanthropic organizations in Singapore, Malaysia, Thailand, and Indonesia		Public-private collaboration in funding of RRT may enable more patients to be treated. Involvement of private organizations in treating ESRD also may engage community participation, promote volunteerism and caring attitudes among the populace, and serve as an avenue for creating awareness of CKD among the community	Implementation of policies that acknowledge and support private contributions, as well as liberal implementation of health care laws, encourages voluntary organizations to participate in the provision of healthcare.
**Bopp (2008)**		Present practical aspects of intervention planning, implementation and evaluation within common community settings				There is a need for process evaluation of intervention implementation to provide valuable information for the dissemination and sustainability of successful interventions	Partnering with community settings (schools, worksites, faith-based organizations and healthcare organizations) offers many benefits and the opportunity to reach specific populations
**DeHaven (2004)**	USA	Examine published literature on health programs in faith-based organizations to determine the effectiveness of these programs	Systematic literature review	Faith-based organizations		Recommendations for FBOs to contribute to community health include increase collaboration between FBOs and health professionals for the purpose of evaluating health activities and disseminating findings, place more emphasis on effectiveness studies as opposed to efficacy studies, devote more attention to building relationships with the racially and ethnically diverse populations that increasingly characterize communities in the United States	Faith-based programs can improve health outcomes
**Theme 4: Establishing a Community of Trust**							
**Maynard (2017)**		Critique the scope and value of recent studies with a focus on obesity-related health promotion in faith organizations	Narrative review	Faith-based organizations, particularly those with African American population		Faith organizations show promise as settings for obesity prevention among high-risk groups, particularly African Americans	Faith organizations are neighborhood focal points and attendance at places of worship remains important for a range of communities
**Vitillo (2017)**		Highlight the role and contributions of select faith-based organizations and some private sector and philanthropic partners, as well as the work of other organizations	Review	Nonstate players and organizations, faith-based organizations, and private sector and philanthropic partners		Lessons learned from the Global Plan in harnessing the strengths of nonstate partners are the ones that should be replicated, enhanced, and taken to scale	Joint engagement and action from state and nonstate players, under the leadership of governments is needed to achieve success within the community
**Lancaster (2014)**	USA	Examine whether using deep structure concepts to increase the salience of health messages in faith-based interventions facilitated changes in weight and related behaviors	Systematic review	African American faith-based organizations	Studies in which the intervention was faith based (incorporating faith-related activities and themes) or faith-placed (taking place in a faith setting), where at least 90% of study participants identified as African American/ Black	Interventions in African American FBOs can successfully improve weight and related behaviors	Faith-based organizations are effective venues for delivering health messages and promoting adoption of healthy behaviors due to the importance of faith to many African Americans
**Bopp (2013)**	USA	Examine current issues associated with the health, behaviors, and well-being of clergy, highlight the literature on the role clergy play in delivering effective health promotion interventions, and present recommendations for improving clergy health and the involvement of clergy in faith-based initiatives	Review	Members of faith-based organizations			Faith-based organizations are led by clergy members who have a strong influence over their institutions and who shape the physical and social environments of their institutions for health-related matters
**Pfefferbaum (2012)**		Describes the application of disaster mental health interventions within the context of the child’s social ecology consisting of the Micro-, Meso-, Exo-, and Macrosystems				In designing services and programs to address both disaster-related trauma and the hardships that disasters create for those in their aftermath, practitioners, providers, and policymakersshould consider the means through which families, peers, teachers and other adults, schools, social programs, services, policy, and other communal systems can assist in minimizing primary and secondary adversities	Individuals, programs, and systems work together through intervention efforts to generate the greatest positive impact. Some of these efforts will emerge naturally within the child’s social ecology. Others may need to be supported by public policies that seek to complement and enrich existing systems
**Francis (2009)**		Review the empirical literature on faith-based HIV prevention programs among African American populations	Literature review	African American faith communities		Several successful faith-based/public health collaborations are identified, and the limitations and strengths of faith-based prevention programs are discussed. Recommendations are provided for developing effective faith-based/public health collaborations	African Americans have close ties to the church and faith-based organizations
**Brooks (2002)**	USA	Explore the relationship between government and faith-based organizations in the opportunities and challenges that are associated with change in this area of health care policy	Review	Government and faith-based organizations		There is a need for further partnering between government and faith-based organizations, within well-defined limits, to maximize the availability of health care education and services throughout this nation	Partnership between government and faith-based organizations could synthesize the best of both worlds and result in a greater health benefit for the nation than either could accomplish alone

**Table 2 vaccines-11-00449-t002:** Summary of Published Studies Focused on the Role of Faith Based Organizations (FBOs) in Vaccination Efforts.

Authors	Origin	Purpose	Research Design	Target Population	Inclusion Criteria	Summary Points	Key Points
Theme 1: Pre-Pandemic Influenza and HPV Vaccination Uptake Efforts							
Kiser & Lovelace (2019)	USA	To assess a national collaboration between the Interhealth Health Program (IHP) at Emory University, the Department of Health and Human Services Partnership Center, the Centers for Disease Control and Prevention, and the Association of State and Territorial Health Officials to prevent the spread of 2009 H1N1 and seasonal influenza by leveraging community organizations (FBOs being one of them)	N/A—no research design was used to execute this vaccination effort	10 sites across the US (was not able to access the table that includes the names of the sites) to serve hard-to-reach vulnerable populations. 5 of the sites were led by faith-based health systems	See previous column	The majority of the faith-based sites conducted vaccination efforts in some capacity—either provided people with vaccinations or conducted immunization education and outreach	The formation of national partnerships is critical in mobilizing local, community resources and organizations such as FBOs to increase vaccination efforts among underserved populations
Bond et al. (2013)	USA	To gauge interest and the capacity of FBOs in medically underserved areas in New York City (NYC) to address health problems among congregation members, focusing on influenza vaccinations	Telephone survey	FBOs in Harlem, the South Bronx, and Central Brooklyn	See previous column	The majority of FBOs showed interest in being an influenza vaccination site	FBOs are equipped with resources that other organizations (academic and government) do not use to deliver health promotion programs to the community. FBO member retention rates are often high because they often have an established connection to the FBO. Thus, making FBOs an effective means of improving community health
Zimmerman et al. (2006)	USA	To assess if interventions tailored to individuals practice sites increased rates of influenza immunization among high-risk children at inner-city health centers of a 2-year period	Pre-post	Children aged 2–17 years with high-risk medical conditions	(1) Child aged 2–17 years (2) Have a high-risk medical condition (3) Be an active patient of the practice (i.e., within the last year)	Faith-based practices saw the highest rates of vaccination and were also associated with the highest likelihood of getting vaccinated.	The use of faith-based health centers was effective in increasing rates of influenza vaccination among high-risk children living in the inner-city
Chebli et al. (2022)	USA	Engage Mexican and Arab communities in Brooklyn to determine various determinants of HPV vaccine uptake	Community-based participatory research	Mexican and Arab parents in Brooklyn, NY	Mexican or Arab heritage, living in Brooklyn, NY	Increasing HPV vaccination rates in Mexican and Arab communities in Brooklyn, NY emphasizes the need for a multilevel, culturally and linguistically relevant program	Importance of utilizing and leveraging assets in the community—including faith-based leaders
Olagoke et al. (2022)	USA	Assess the associations between 3 domains of religiosity (organizational, non-organizational, and intrinsic) and the intention to obtain HPV information and receive the HPV vaccine	Cross-sectional	Christian parents across the USA	Parents must be: (1) Over 18 years at recruitment; (2) Parent/guardian of at least 1 child 11–17 who has never received the HPV vaccine; (3) Christian; (4) Live in the USA	Study findings showed a positive association between the organizational domain and the intention to seek HPV information. Information-seeking may not lead to vaccination, however	Importance of engaging FBOs and empowering them to provide parents with accurate information regarding HPV vaccination in order to increase HPV vaccination rates
**Theme 2: Value of Addressing Vaccine Disparities in Ethnic Minority Communities**							
Santibañez et al. (2019)	USA	Discusses how the collaboration of the CDC with FBOs played a key role in the response to pandemic influenza (2009), Ebola (2014), and Zika (2016). Note: vaccination efforts will be made in reference to pandemic influenza (2009)	N/A—no research design was used to execute this vaccination effort	African American, African-born, Asian/Pacific Islander, Hispanic/Latino, and American Indian communities in the Minneapolis-St. Paul area	See previous column	The Minnesota Immunization Networking Initiative (MINI) conducted vaccination clinics at various places of worship—churches, a Hindu Temple, mosques, a Buddhist monastery, etc., and provided free influenza vaccinations	The collaboration of MINI with various FBOs to provide underserved communities with influenza vaccinations helped address barriers to vaccination (access, mistrust, transportation, etc.)
Daniels et al. (2007)	USA	To examine (1) whether church-based vaccine education increases adult utilization of vaccinations in ethnic/minority communities and (2) if churchgoers who are offered vaccinations in churches have higher vaccination utilization rates compared to non-churchgoers	Randomized Controlled Trial	African American and Latino adults aged 65 and older in the Bay Area (San Francisco, CA, USA)	(1) Previously unvaccinated with pneumococcal vaccine (2) Not regularly receiving influeza vaccine (3) 65 years or older or clinical indication for vaccination (various chronic diseases)	The intervention showed that vaccinations provided by FBOs increased vaccination rates among racial/ethnic groups. Additionally, the provision of onsite vaccinations at FBOs increased vaccination rates compared to programs that focus only on education	FBOs may play an important in decreasing the disparities in vaccination rates among racial/ethnic communities.
**Theme 3: Addressing Recent COVID-19 Vaccination Efforts**							
Rachmawati Umniyatun Rosyidi Nurmansyah (2022)	Indonesia	To assess the strengths of Indonesia’s two largest Islamic faith-based organizations (FBOs) and the challenges faced while conducting activities to mitigate the impact of COVID-19 in Indonesia	Qualitative Case Study	Informants—the heads of the special units of both FBOs, government officials, the FBOs’ charity business managers, and community members who benefitted from the FBOs’ efforts	Have knowledge and experience to contribute to study objectives (various informants)	Indonesia’s two largest Islamic FBOs have strengths such as their organizational structure to target the grassroots and are able to support the programs of other organizations that engage in primary and secondary prevention efforts to mitigate the impact of COVID-19. Importantly, the Central Board of one of the FBOs provided support for COVID-19 vaccination implementation	The collaboration of FBOs with the government aids in the mobilization of resources to help reduce the impact of COVID-19.
Monson et al. (2021)	USA	An academic hospital to engage FBOs through meetings to communicate information about the COVID-19 pandemic and the role they can play in mitigating the impact of the pandemic on communities	Narrative review	FBOs in Baltimore City and the Maryland area (other states such as Michigan, Pennsylvania, Georgia, and New York also participated)	No specific inclusion criteria—findings were based on participating FBOs	The meetings with FBOs included COVID-19 related themes on how the FBOs can communicate public health messaging, serving as a COVID-19 testing site, and as potential COVID-19 vaccine sites	Medical-religious partnerships are practical and valuable in playing an important role in minimizing the impact of COVID-19 related disparities.
Dascalu et al. (2021)	Romania	To discuss the contributions of the Romanian Orthodox Church (ROC) in mitigating the impact of COVID-19; vaccination efforts being one	Case study	Followers of the ROC	See previous column	The ROC actively advocated against misinformation regarding COVID-19 in general and the national vaccine campaign	FBOs, as a result of partnering with governmental agencies may play a key role in mitigating COVID-19 outcomes by advocating for the national vaccine campaign

## Data Availability

No new data were created or analyzed in this study. Data sharing is not applicable to this article.

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
