# Peer review of "The Role of Faith-Based Organizations in Improving Vaccination Confidence & Addressing Vaccination Disparities to Help Improve Vaccine Uptake: A Systematic Review"

_vaccines, 2023, doi:10.3390/vaccines11020449_

Round 1
Reviewer 1 Report
This manuscript was described the relationship between vaccination and faith-based organizations as a systematic review. For vaccine preventable diseases, it is important to maintain high vaccination rates. However, vaccination have been often prevented by misinformation, distrust, and hesitancy regarding the uptake of vaccine. It was concluded that FBOs play the important role of vaccination. My comments are following.
1. Table 1 and 2 are not presented.
2. The relationship between FBOs and believers may be different between countries or kinds of religions. In countries with weak relationship for FBOs, such as Chinese or Japanese, FBOs may not contribute for vaccination. Also, FBOs in Korea were aggravated the COVID-19 outbreak by spreading fake news. In this review, although 14 of the 26 articles were conducted in the United States and I understand as the results in USA mainly, 12 of the 26 articles were results in other countries. Readers want to know the role of FBOs in each country about vaccination. Please comment the differences in role and characters of FBOs between countries or religions in Discussion.
3. L211: Superscript “7” will be “[7]”.
Author Response
Reviewer 1
Comment: Table 1 and 2 are not presented.
Response: We apologize for this oversight on the original submission. All tables previously references are now included in the revision.
Comment: The relationship between FBOs and believers may be different between countries or kinds of religions. In countries with weak relationship for FBOs, such as Chinese or Japanese, FBOs may not contribute for vaccination. Also, FBOs in Korea were aggravated the COVID-19 outbreak by spreading fake news. In this review, although 14 of the 26 articles were conducted in the United States and I understand as the results in USA mainly, 12 of the 26 articles were results in other countries. Readers want to know the role of FBOs in each country about vaccination. Please comment the differences in role and characters of FBOs between countries or religions in Discussion.
Response:
We thank the reviewer for this important point and made minor changes throughout to ensure clarity as well as included the following paragraphs within the discussion.
Notably, we also found that the majority of research exploring the role of FBOs and faith-based engagement, occurred in the US (14 of 26), other countries examined came up singularly such as: Barbados, Ethiopia, Germany, Ghana, Israel, Indonesia, Netherlands, Romania, and Southeast Asia broadly. Given the disproportionate volume of research focused on efforts in the US, comparing findings across countries based on available evidence is challenging. However, consistently, while the value of FBOs and faith-based engage-ment is identified particularly in supporting underserved and traditionally marginal-ized populations, most studies underscored the importance of coordination with on-going vaccination efforts so as to not undermine or duplicated efforts, but rather en-hance and build upon them. No discernable differences were identified between how FBOs are engaged across countries included in the studies that were captured in this review, with the exception of Rujis et al.[55] which focused on the role of Protestant re-ligious leaders in the Netherlands, which found that in this case, engaging Protestant religious leaders to support vaccination efforts may be ineffective as leaders were un-willing to promote vaccination and there was a low level of religiosity in the general population. This provides important insights moving forward in that the potential impact of FBOs and faith-based engagement in supporting vaccination efforts may be determinant on the local level faith-community’s buy-in and willingness to engage on this issue as well as the level of religiosity in the community.
We further acknowledged the disproportionate representation in studies in the limitations and opportunities for future research:
Future research should also further examine any differences or commonalities in the role of FBO and faith-based engagement across high income and low- and mid-dle-income countries.
Comment: L211: Superscript “7” will be “[7]”.
Response: We apologize for the oversight on the original submission. The superscript “7” has been fixed to “[7]”.
Reviewer 2 Report
The paper is interesting but it has to be improved.
Major Issues
In the paper, the authors refer to several tables (Table 1, Table 2, and Table 3), but there are no tables. The manuscript cannot be evaluated deeply until the tables are included.
Minor issues.
Please provide examples of what a FOB is. Is a church a FOB? Are Red Crescent or Magen David Adom FBOs? Is Oxfam, founded by Quakers, a FOB?
In lines 97-98, the authors excluded "non-peer-reviewed or non-research papers." It is clear that Preprints were excluded. The question is that there are journals that do not have peer review. How did the authors know that journals were peer-reviewed? Did they check each journal's author instructions or consult a list of peer-reviewed journals?
In the material and methods, the authors should write what persons did the bibliographic search and who extracted de data from the papers.
Please cite the "Quality Rating Scheme of the Oxford Centre for Evidence-Based Medicine."
Please check the links in the bibliography and write the date of consultation. The link of reference number 14 https://www.nhlbi.nih.gov/health-topics/study-quality-assessmenttools did not work.
.the actual link is https://www.nhlbi.nih.gov/health-topics/study-quality-assessment-tools (accessed on 31-01-2023)
The authors used the National Institutes of Health (NIH) rating system for detecting bias. There are several instruments. What instruments did they use? What was the evaluation for each study
Author Response
Reviewer 2
Comment: In the paper, the authors refer to several tables (Table 1, Table 2, and Table 3), but there are no tables. The manuscript cannot be evaluated deeply until the tables are included.
Response: We apologize for this oversight on the original submission. All tables previously references are now included in the revision.
Comment: Please provide examples of what a FOB is. Is a church a FOB? Are Red Crescent or Magen David Adom FBOs? Is Oxfam, founded by Quakers, a FOB?
Response:
Thank you for elevating this. We offer some clarity in the introduction building off the initial definition we included. Newly added language is italicized below.
FBOs and faith-based engagement strategies have been the foundation of many previ-ous collective efforts targeting other infectious diseases and public health emergen-cies.[6,7,13] FBOs are organizations whose philosophies are driven by certain religious beliefs, often including a social or moral component.[14] These entities have been shown to bring people together for positive purposes and can present powerful agents for health and justice.[15–18] As religion is a social determinant of population health, it functions through the work of social institutions.[16] Therefore, FBOs are players in communities; they present a discernible, public face to communities through acts of leadership and capacity for service to others.[15,17] The CDC workbook defined FBOs as “churches, synagogues, mosques, church spon-sored service agencies, and all charitable organizations with religious affiliations” – this broad definition can therefore include nonprofit organizations with a religious af-filiation or inspiration.[18]
Comment: In lines 97-98, the authors excluded "non-peer-reviewed or non-research papers." It is clear that Preprints were excluded. The question is that there are journals that do not have peer review. How did the authors know that journals were peer-reviewed? Did they check each journal's author instructions or consult a list of peer-reviewed journals?
Response: Thank you for pointing this out. Based on this comment, we have removed the exclusion criteria of "non-peer-reviewed or non-research papers." This revision can be found on line 97.
Comment: In the material and methods, the authors should write what persons did the bibliographic search and who extracted de data from the papers.
Response: Thank you for this suggestion. We have included the more details of the data extraction process. This revision can be found in lines 120-122.
Comment: Please cite the "Quality Rating Scheme of the Oxford Centre for Evidence-Based Medicine."
Response: Thank you for pointing this out. Based on this comment, we have added the citation for the "Quality Rating Scheme of the Oxford Centre for Evidence-Based Medicine." Please see lines 481-482.
Comment: Please check the links in the bibliography and write the date of consultation. The link of reference number 14 https://www.nhlbi.nih.gov/health-topics/study-quality-assessmenttools did not work.
Response: Thank you for pointing this out. Based on this comment, we have updated the correct link. Please see lines 484.
Comment: The authors used the National Institutes of Health (NIH) rating system for detecting bias. There are several instruments. What instruments did they use? What was the evaluation for each study?
Response: Based on this helpful comment, we have incorporated the information on the details of the tools used to review bias. This revision can be found on lines 124-125.
Reviewer 3 Report
I was invited to revite the paper entitled "The Role of Faith-Based Organizations in Improve Vaccination Confidence & Addressing Vaccination Disparities: A Systematic Review". It was a systematicreview that aimed to evaluate the role of FBOs in improving childhood vaccination.
The topic is interesting and results were clearly presented.
Despite that, Authors should improve the paper:
- As supplementary material, Authors should report the exact research strategy used for all databases;
- Included papers should be also presented in a Table 1, reporting all studies characteristics;
- Risk of bias assessment was not reported;
- Among discussions, half of included papers were from US, so Authors should better descibe differences among countries.
Author Response
Reviewer 3
Comment: As supplementary material, Authors should report the exact research strategy used for all databases.
Response: Based on this helpful comment, we have incorporated supplementary material to describe research strategy.
Comment: Included papers should be also presented in a Table 1, reporting all studies characteristics.
Response: We apologize for this oversight on the original submission. All tables previously references are now included in the revision.
Comment: Risk of bias assessment was not reported.
Response: Based on this helpful comment, we have incorporated the information on the details of the tools used to review bias. This revision can be found on lines 124-125.
Comment: Among discussions, half of included papers were from US, so Authors should better describe differences among countries.
Response:
We thank the reviewer for this important point and made minor changes throughout to ensure clarity as well as included the following paragraphs within the discussion.
Notably, we also found that the majority of research exploring the role of FBOs and faith-based engagement, occurred in the US (14 of 26), other countries examined came up singularly such as: Barbados, Ethiopia, Germany, Ghana, Israel, Indonesia, Netherlands, Romania, and Southeast Asia broadly. Given the disproportionate volume of research focused on efforts in the US, comparing findings across countries based on available evidence is challenging. However, consistently, while the value of FBOs and faith-based engage-ment is identified particularly in supporting underserved and traditionally marginal-ized populations, most studies underscored the importance of coordination with on-going vaccination efforts so as to not undermine or duplicated efforts, but rather en-hance and build upon them. No discernable differences were identified between how FBOs are engaged across countries included in the studies that were captured in this review, with the exception of Rujis et al.[55] which focused on the role of Protestant re-ligious leaders in the Netherlands, which found that in this case, engaging Protestant religious leaders to support vaccination efforts may be ineffective as leaders were un-willing to promote vaccination and there was a low level of religiosity in the general population. This provides important insights moving forward in that the potential impact of FBOs and faith-based engagement in supporting vaccination efforts may be determinant on the local level faith-community’s buy-in and willingness to engage on this issue as well as the level of religiosity in the community.
We further acknowledged the disproportionate representation in studies in the limitations and opportunities for future research:
Future research should also further examine any differences or commonalities in the role of FBO and faith-based engagement across high income and low- and mid-dle-income countries.
Round 2
Reviewer 2 Report
The authors have incorporated, satisfactorily into their paper, all the critics of the first revision.
Reviewer 3 Report
All comments were properly addressed. The paper can be accepted for publication.